# Pre-Harvest Food Safety Challenges in Food-Animal Production in Low- and Middle-Income Countries

**DOI:** 10.3390/ani14050786

**Published:** 2024-03-02

**Authors:** Eyasu T. Seyoum, Tadesse Eguale, Ihab Habib, Celso J. B. Oliveira, Daniel F. M. Monte, Baowei Yang, Wondwossen A. Gebreyes, Walid Q. Alali

**Affiliations:** 1Ohio State Global One Health, Addis Ababa 62347, Ethiopia; seyoum.3@osu.edu (E.T.S.); habtemariam.2@osu.edu (T.E.); celso.bruno.oliveira@gmail.com (C.J.B.O.); gebreyes.1@osu.edu (W.A.G.); 2Department of Veterinary Medicine, College of Agriculture and Veterinary Medicine, United Arab Emirates University, Al Ain P.O. Box 15551, United Arab Emirates; i.habib@uaeu.ac.ae; 3Department of Environmental Health, High Institute of Public Health, Alexandria University, Alexandria P.O. Box 21511, Egypt; 4ASPIRE Research Institute for Food Security in the Drylands (ARIFSID), United Arab Emirates University, Al Ain P.O. Box 15551, United Arab Emirates; 5Department of Animal Science, College for Agricultural Sciences, Federal University of Paraiba (CCA/UFPB), Areia 58397-000, PB, Brazil; monte_dfm@alumni.usp.br; 6College of Food Science and Engineering, Northwest A&F University, Yangling, Xianyang 712100, China; ybw090925@163.com; 7Department of Veterinary Preventive Medicine, The Ohio State University, Columbus, OH 43214, USA; 8Department of Biostatistics & Epidemiology, College of Public Health, East Tennessee State University, Johnson City, TN 37614, USA

**Keywords:** food safety, pre-harvest, LMICs, interventions, AMR

## Abstract

**Simple Summary:**

Consuming unsafe food globally results in millions of illnesses and deaths. The safety of our food, from farm (pre-harvest) to table (postharvest), is paramount. In low- and middle-income countries (LMICs), ensuring food safety is challenging. This review explores current insights into pre-harvest critical issues related to food safety. At the farm level, critical food safety issues include animal health, antimicrobial resistance, and farmers’ knowledge, attitudes, and practices. Globally, several pre-harvest interventions aim to mitigate risks from animal-origin food. Despite effectiveness at a small scale, these interventions often lack scalability. Many countries have established laws and regulations to shield consumers from unsafe food consumption. However, food safety regulations in several LMICs are weak, inconsistent, and inadequately enforced. Preventing foodborne diseases requires a One Health approach, uniting experts in animal, human, and environmental health. Despite its importance, the application of this approach to address food safety issues is limited in low- and middle-income countries.

**Abstract:**

Food safety remains a significant global public health concern, with the risk of unsafe food varying worldwide. The economies of several low- and middle-income countries (LMICs) heavily rely on livestock, posing a challenge to ensuring the production of safe food. This review discusses our understanding of pre-harvest critical issues related to food safety in LMICs, specifically focusing on animal-derived food. In LMICs, food safety regulations are weak and inadequately enforced, primarily concentrating on the formal market despite a substantial portion of the food sector being dominated by informal markets. Key critical issues at the farm level include animal health, a low level of good agriculture practices, and the misuse of antimicrobials. Effectively addressing foodborne diseases requires a comprehensive One Health framework. Unfortunately, the application of the One Health approach to tackle food safety issues is notably limited in LMICs. In conclusion, considering that most animal-source foods from LMICs are marketed through informal channels, food safety legislation and policies need to account for this context. Interventions aimed at reducing foodborne bacterial pathogens at the farm level should be scalable, and there should be strong advocacy for the proper implementation of pre-harvest interventions through a One Health approach.

## 1. Introduction

The consumption of unsafe food has severe global implications, leading to millions of sicknesses and thousands of deaths annually [1]. According to the World Health Organization (WHO), the global impact includes the loss of 33 million years of healthy lives each year [2]. Contaminated food products worldwide contribute to an estimated six hundred million cases of illness, 420,000 deaths, and economic losses exceeding 100 billion USD [3]. The risk of disease due to unsafe food varies across countries. In the USA, an estimated 47.8 million cases of annual foodborne illness occur [4]. The Middle East and North Africa (MENA) regions rank third globally in the estimated burden of foodborne diseases per capita, with 70% attributed to pathogens like *E. coli*, non-typhoidal *Salmonella* (NTS), *Campylobacter*, and *Norovirus* [2]. This underscores the significant impact of these agents on foodborne infections in the region. China reports 19,517 foodborne outbreaks from 2003 to 2017, resulting in 235,754 illnesses, 107,470 hospitalizations, and 1,457 deaths, with bacterial pathogens accounting for about 40% of the outbreaks [5]. Specifically, *Salmonella* spp., *Campylobacter* spp., *Shigella* spp., *Cryptosporidium* spp., and Shiga toxin-producing *Escherichia coli* O157 are linked to approximately 500,000 illnesses and 12,000 deaths annually [6]. These figures emphasize the urgent need for global efforts to address and prevent the impact of unsafe food consumption on public health. Unfortunately, there is a significant lack of empirical information regarding the incidence of foodborne illness in many LMICs [7]. Even with existing information, a critical concern is the unreliability of data sources, often marked by underreporting and a narrow focus on a single food hazard. This implies that the majority of foodborne illnesses in LMICs, especially in most African countries, go unreported, and the true disease burden remains undetermined.

In high-income countries (HICs), communities generally possess more knowledge about the potential risks of unsafe food, leading to a continuous demand for safer and higher-quality food. In response, governments in HICs actively work to address food safety concerns by establishing legislative and regulatory systems. In contrast, the situation in developing countries is starkly different, especially considering that the primary tragedy of foodborne illnesses occurs there. While the full extent of the foodborne-disease problem in LMICs remains unknown, it is widely believed that these countries are disproportionately affected [8]. Factors linked to the risk of foodborne disease in LMICs encompass various issues, including the lack of adequate food storage infrastructure, poor food production practices marked by the inappropriate use of agricultural chemicals, the use of unsafe water for cleaning and processing food, and inadequacies or inconsistencies in enforcing regulatory standards [2,8]. The World Health Organization (WHO) defines food safety as the state wherein there is assurance that food will not adversely affect the consumer’s health when prepared and/or consumed following its intended usage [9]. This definition emphasizes the importance of proper conditions and precautions throughout the entire food preparation, manufacturing, processing, storage, and distribution process to ensure that food is wholesome, safe, and suitable for human consumption [9]. 

The economy of LMICs, especially in rural areas, heavily relies on livestock. Food producers and various stakeholders along the value chains in LMICs depend significantly on the livestock sector for income generation and job creation, bearing substantial economic implications. This dependency is crucial for impoverished communities, particularly women and pastoralists, where livestock serves as a vital resource and safety net for their livelihoods. Moreover, beyond economic aspects, livestock represents a substantial source of food for billions of people in both rural and urban areas [10]. The livestock sector is the fastest-growing agricultural subsector in LMICs, playing a vital role in transforming economic and social growth in these countries [11]. Close to a billion smallholder farmers in LMICs support their livelihood through engagement in the livestock production system [11,12].

For small-scale farmers in developing countries, owning livestock is not just an important asset for income but also a primary source of animal-origin food. Livestock plays multifaceted roles, serving as agricultural inputs, providing traction, and functioning as a safety net during emergencies or lean agricultural seasons. In this context, livestock represents a versatile and indispensable resource that significantly contributes to the livelihoods and sustenance of these farmers. In addition to supporting livelihoods, the livestock sector plays a crucial role in ensuring food and nutrition security for nearly 1.3 billion smallholders. Moreover, the contribution of the livestock sector to the global value of agricultural output from developing countries is substantial, reaching up to 40% [13].

Globally, there are myriad methods for raising livestock, resulting in a diverse array of goods and services. These methods leverage different animal species and resource combinations within various agro-ecological and socio-economic contexts. To categorize this diversity, various livestock production systems (LPS) have been identified based on distinctive patterns observed in livestock production. A common classification revolves around how animals utilize the land, leading to widely accepted distinctions between grazing systems, mixed farming systems, and industrial (or landless) systems [14]. This categorization reflects the predominant ways in which livestock interact with their environments and the prevailing agricultural practices in different regions, providing insights into the varied and dynamic nature of livestock production worldwide.

Food safety concerns are broad, encompassing physical, biological, and chemical hazards on a global scale. While these hazards can manifest at any stage in the food value chain, issues arising at the pre-harvest level are particularly significant. This review centers on LMICs, with examples from a number of countries/regions (i.e., China, Ethiopia, Brazil, and the Middle East). Moreover, the review has a specific emphasis on food of animal origin, to delve into the current understanding of pre-harvest critical issues related to foodborne disease caused by bacterial pathogens and food safety in general. Through this exploration, the goal is to shed light on the implications of these issues for overall food safety in LMICs.

## 2. Overview of Food Safety Legislative and Regulatory Frameworks and Implementation Challenges in LMIC

Food safety legislation comprises a collection of acts, regulations, requirements, or procedures relevant to foodstuffs enacted by government authorities with the intention of safeguarding public health [15]. Once legislators pass regulations, their ground-level implementation, including enforcement, occurs through regulatory frameworks that constitute food safety regulatory and enforcement bodies [16]. Different countries employ various mechanisms to determine the structural framework of their food safety regulatory systems. For instance, at the regional level in the Middle East, the Gulf Cooperation Council (GCC) Standardization Organization (GSO) plays a pivotal role in enhancing food safety and promoting trade by establishing standardized regulations. The GSO facilitates harmonization among GCC member states, ensuring a consistent and transparent framework for food safety practices. This harmonization, notably at the regional level, is vital for promoting cross-border trade, reducing barriers, and fostering economic cooperation in the region, ultimately benefiting consumers and promoting regional economic growth.

The establishment of an effective food safety system necessitates the implementation of a national legal framework. Food is governed by a complex web of laws and rules in every nation, outlining the standards that food chain operators must meet to ensure food safety and quality. The institution of such laws and regulations involves specifying the details of how food is produced, traded, and handled. Consequently, it encompasses all relevant aspects of food commerce along the entire food supply chain, from the provision of animal feed to the consumer, and regulates food control, quality, safety, and relevant aspects of trade [17].

In many developing countries in Africa and Asia, some form of legislation and regulatory laws related to food safety have been enacted. However, the way the food system operates in LMICs inherently contributes to the complexity of observed food safety governance challenges in these countries. For instance, in China, to prevent farmers from unsafely selling dead pigs, the government has implemented two main initiatives. The first is the supervision and punishment policy, which involves a four-level supervision system comprising province, city, county, and township levels. At the county level, collaboration among livestock, health, forestry, and market supervision departments forms an enforcement team overseeing key areas and populations. Grid management is implemented in designated areas, and once illegal acts are discovered, administrative penalties are applied in accordance with relevant laws. The second initiative is the subsidy policy, where farmers are subsidized 80 yuan per head if dead pigs are harmlessly disposed of through methods such as deep burial and incineration [18]. Furthermore, in China, the government has introduced several plans to enhance food safety, including the “Food Safety Law of the People’s Republic of China”, the “Animal Epidemic Prevention Law of the People’s Republic of China”, the “Regulations on the Management of Pig Slaughtering”, and the “Three Year Action Plan for Strict Standards to Promote and Improve Safety in Livestock and Poultry Slaughtering” [19,20,21,22]. These plans aim to regulate meat safety across various stages, from animal breeding and slaughter to distribution.

It’s essential to note that these regulations and measures represent only a portion of the comprehensive policy framework. The Chinese government has developed numerous other regulations and standards related to food safety and animal husbandry management. Since policy documents may undergo updates or adjustments, it is crucial to refer to the latest government documents and regulations to gain an accurate understanding of the current regulatory content.

Food safety legislative and regulatory frameworks in the Middle East play a crucial role in ensuring the well-being of the population. However, several challenges persist in their effective implementation. The region faces complexities due to diverse cultural practices, varying levels of economic development, and a multitude of stakeholders involved in the food supply chain. Harmonizing and enforcing consistent standards across Middle Eastern countries remains a challenge, as each nation often has its own set of regulations. In many cases, inadequate infrastructure and resources hinder the proper monitoring and inspection of food production and distribution. There is a clear need for capacity building and training programs for regulatory agencies to enhance their capabilities in risk assessment and management. Additionally, addressing issues such as cross-border food trade and international collaboration in sharing best practices are essential for establishing an effective regional food safety system [23,24].

A survey conducted in nine African countries has revealed that food safety regulatory measures and associated infrastructures are mostly absent. Even in countries that have some sort of regulatory framework, they are often not well-suited for the country’s context, where most markets are informal in nature. These regulations are essentially derived from developed countries. The situation analysis of food safety legislation conducted in the nine African countries, spanning eastern, western, and southern parts of the continent, revealed that legislative acts involve multiple institutes and exhibit characteristics of being overlapping, unfocused, and generic in nature. In general, existing laws and regulations from multiple organizations do not effectively target all the different stages and components in the food value chain [25]. Legislative acts or policies give overlapping mandates to different institutes, often resulting in confusion about who should be accountable for certain aspects of food safety issues. As an example, in Ethiopia, three ministerial organizations, namely, the Ministry of Health, Ministry of Agriculture, and Ministry of Trade, are legally mandated to implement food safety policies in their respective institutes (illustrated in Figure 1). However, the country lacks laws and regulations that require these three organizations to educate the public on food safety hazards, set food standards, and implement a food code. Additionally, the regulatory framework in Ethiopia does not adequately address food safety issues arising from informal markets [26].

Although comprehensive and representative data on food safety perception in LMICs is lacking, the concept of food safety in these countries appears to have gained increased attention from consumers. This shift in behavior is evident in various instances during foodborne outbreaks, where people have exhibited changes ranging from entirely avoiding implicated food items to actively questioning the eligibility of food vendors for service provision. In China, for example, public concern about the safety of the food they consume has escalated to the point where it is considered a top safety priority on a daily basis, ranking closely with natural disasters such as earthquakes [8]. Notably, women express more apprehension about food safety than men, and urban residents exhibit greater concerns compared to their rural counterparts. Additionally, significant variations in food safety perceptions exist across different age groups; respondents under the age of 31 demonstrate heightened concerns, while those over 44 express the least worry about food safety. Furthermore, distinct differences in food safety perceptions exist between netizens and non-netizens. Intriguingly, across all sub-samples, netizens consistently exhibit greater concerns about food safety than non-netizens [27]. Such growing concern naturally puts pressure on governments in LMICs to prioritize these issues.

Food safety concerns in LMICs hold economic implications, particularly in nations where exports of animal and animal products form the economic backbone. In LMICs, close to 1 billion smallholder livestock producers contribute 40% of the agricultural gross domestic product and from 2% to over 33% of household incomes, playing a significant role in the safety of foods of animal origin [12]. As expected, there is a tendency in LMICs to adopt international requirements such as Codex and ISO standards, especially for processed foods. These measures are anticipated to enhance consumers’ trust in the supervision and sampling inspection of commonly consumed foods. Key factors contributing to increased trust include ensuring that the sampling process is open and transparent, implementing rigorous regulations, and establishing the most stringent standards [28]. However, these legal frameworks are frequently not fully implemented as intended. There is a lack of comprehensive enforcement of regulations, with a predominant focus on the formal market, while a significantly larger portion of the food sector is dominated by the informal market. Food vendors who sell their products in markets regulated by government bodies are categorized as operating within the formal sector, whereas those selling products in places not regulated by government bodies are considered to be part of the informal sector [29]. Additionally, despite their nominal existence, food safety systems in LMICs are characterized by being weak, inconsistent, and ineffective in protecting the general public from food hazards. Moreover, the existing system lacks the capability to make countries competitive enough in the global food market [30].

Undoubtedly, enhancing the safety of food entails additional costs, which, unfortunately, many developing countries often find challenging to invest in. Some experts contend that investments made in food safety over the past two decades have not yielded the required impact commensurate with the resources allocated. The perceived lack of impact does not stem from the ineffectiveness of the strategies employed but rather underscores the importance of considering other factors such as globalization, shifts in eating habits, and changes in farming practices [31].

**Figure 1 animals-14-00786-f001:**
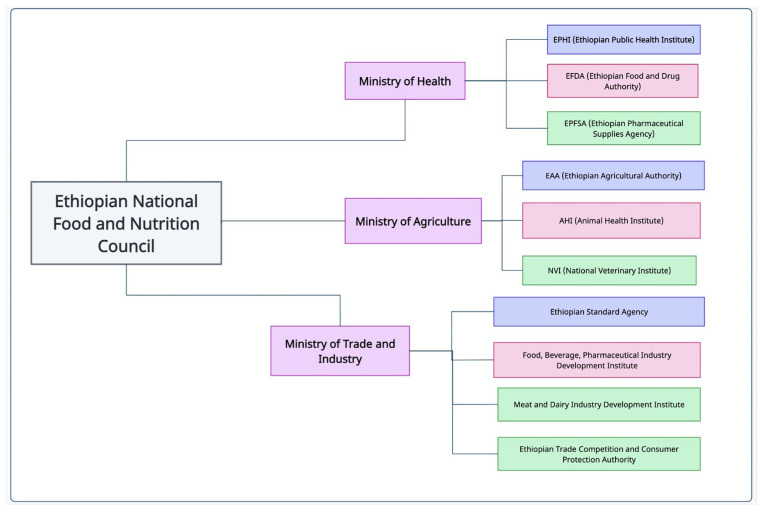
Example of three main line ministries with a legal mandate on food safety in Ethiopia [32].

## 3. Critical Food Safety Issues at Farm Level in LMIC 

### 3.1. Animal Health: Implication on Food Safety

The health status of food animals significantly influences the likelihood of foodborne disease occurrence. Generally, healthier animals pose a reduced risk of shedding bacterial foodborne pathogens, thereby lowering the potential for the contamination of animal products. For instance, healthier animals may shed lower levels of bacterial foodborne pathogens; however, the risk of microbial contamination and cross-contamination can increase due to handling infected animals that are carriers of higher loads of pathogens. Hence, gastrointestinal ruptures during animal processing can cause contamination of the meat.

Implementation of pre-harvest measures to enhance animal health and reduce the risk of bacterial infection is important to improve food safety along the food chain. Such measures include good animal husbandry practices, including providing safe and hygienic animal feed, stress reduction in handling animals, and implementing robust biosecurity practices. Strong biosecurity practices on farms are highly important to prevent and control diseases of food animals and to safeguard the public from foodborne pathogens and other hazards. The adoption and implementation of national biosecurity standards, along with associated frameworks, laws, policies, and strategies, are, however, minimal in LMICs. A recent qualitative study from Kenya revealed limited adherence to biosecurity practices due to factors such as personal preferences, constraints in veterinary service delivery, enforcement limitations, and inadequate infrastructure [33]. Additionally, the study highlighted several preventive actions at the farm level, including the management of buildings and facilities where animals are kept, attention to surroundings and environmental control, practices related to animal feeding, watering, farm management, preparation of animals for slaughter, and common measures for record-keeping and traceability. Moreover, enhancing animal welfare including keeping animals clean improves animal health and reduces the risk of bacterial infection. These actions are identified as crucial in mitigating the risk of food contamination at the farm level [34].

Disinfection is the most common method of sterilization in slaughterhouses. It serves to eliminate pathogenic microorganisms from the environment, curbing cross-contamination and enhancing the safety of animal-origin food to a certain extent. Overall, disinfection procedures play an important role in controlling environmental contamination and minimizing the risk of meat contamination by pathogenic bacteria [35].

Prohibiting the unsafe sale of livestock that have died in production and harmlessly disposing of dead animals are key measures to control and prevent outbreaks of bacterial zoonotic diseases. These measures hold great significance for maintaining the safety of meat-derived food and public health [18].

Antibiotics play a critical role in treating animals affected by bacterial pathogens. Severe bacterial infections in animal populations are often controlled by administering appropriate antibiotics. The rational use of antibiotics ensures good animal health and welfare, positively impacting food safety, preventing disease outbreaks, and saving lives as well as the livelihoods of farmers [36]. However, one major drawback is the misuse and overuse of antibiotics for treatment, prophylactic, and growth promotion purposes. This misuse could lead to the emergence and persistence of antibiotic-resistant populations of bacteria on the farm and the spread of antibiotic-resistant organisms and resistance genetic markers to humans through direct contact or the consumption of food of animal origin.

In LMICs, the consumption of animal-source foods, such as meat and milk, is relatively very low. For instance, in Ethiopia, despite having one of the largest livestock populations globally, the average per capita annual consumption of meat, eggs, and dairy products is notably low at 4.6 kg, 0.2 kg, and 16.7 kg, respectively. These figures are considerably lower, amounting to less than one-tenth of consumption levels observed in the U.S. [37]. In China, the average consumption of meat, especially pork, has continued to increase. Pork consumption increased from 37.1 g/d in 1992 to 64.3 g/d in 2012. The average consumption of dairy products increased from 14.9 g/d in 1992 to 24.7 g/d in 2012 [38]. From 1987 to 2020, the consumption of animal-based food, such as meat, eggs, and milk, has continued to increase. Per capita consumption of eggs rose by 3.3 times, from 3.9 kg in 1987 to 12.7 kg in 2020, but it remained below the minimum intake in the dietary guidelines. The per capita milk consumption rose from 1.9 kg in 1987 to 12.1 kg in 2020, which was only 11.1% of the milk intake in the dietary guidelines. Per capita consumption of livestock and poultry meat rose from 22.5 kg in 1987 to 61.1 kg in 2020, increasing 2.7 times in 33 years [39].

Using traceability throughout the production chain, from pre-harvest through post-harvest, is crucial for reducing the risk of foodborne diseases. In Europe, the implementation of animal identification and movement databases has informed slaughterhouses about the health status of animals, enabling the classification of animals based on their food safety risk categories [40]. However, such a system is not currently available in LMICs and may not be practically feasible considering the informal nature of the system.

Farm animals in LMICs are known to harbor major foodborne bacterial pathogens that pose public health concerns. Numerous studies conducted in LMICs have identified a variety of bacteria in animals, animal-source foods, and the farm environment. For instance, a recent study in Ethiopia found that *E. coli* O157:H7 was isolated from lactating cow milk and farm environmental samples at rates ranging from 0.2% to 4.7% [41,42,43,44]. Similar findings of various bacterial foodborne pathogens have been reported in other LMICs, with different rates observed in food of animal origin and farm environmental samples (as shown in Table 1).

### 3.2. Antibiotic Resistance as a Food Safety Issue at Farm Level

The discovery of antibiotics is considered one of the major advances in human and animal health. The use of antibiotics in animals has significantly contributed to improving the health and productivity of animals. However, the overuse and misuse of antibiotics in food animals have resulted in the selection of antibiotic-resistant pathogens in these animals [58]. This is particularly common in intensive poultry farms, pig farms, and dairy farms where a large number of animals are kept in crowded conditions. Antibiotics are used in food animals for both therapeutic and prophylactic purposes. The frequent use of antibiotics reduces the proportion of susceptible organisms, enabling resistant strains to prevail [59].

Due to the selective pressure exerted by the overuse of antibiotics in animal husbandry, bacterial progeny cells inherit crucial genome alterations, such as point mutations that promote high-level resistance to critically important antimicrobials. More importantly, these bacteria can also spread drug-resistant genes horizontally via mobile genetic elements, such as plasmids and transposons. This process may accelerate the widespread dissemination of established antibiotic markers at farms, leading to the emergence of resistant infectious bacterial pathogens that are difficult to treat in animals on the farm, causing increased morbidity, mortality, and loss of productivity. However, this event is not limited to farms. The selected resistant population of bacteria spreads to the nearby environment, particularly in farms where vectors and reservoirs, such as flies and migratory birds, disseminate resistant bacteria over some distance [60]. Upon migration, these vectors carry these bacteria to new regions, sometimes spanning across countries or continents, contributing to the global spread of antibiotic resistance. In this regard, Fuentes-Castillo et al. [61] isolated five extended-spectrum beta-lactamase (ESBL)-producing *Escherichia coli* from migratory and resident gulls, indicating the importance of these animals as reservoirs. To mitigate this issue, it is important to manage the waste produced by farmers correctly; otherwise, wildlife, including birds, insects, and small mammals, could be exposed to antimicrobial residues and resistant bacteria [62]. 

Multi-drug resistant (MDR) organisms can spread from food animals to humans through contaminated food. While most foodborne pathogens, such as *Salmonella*, *E. coli* O157:H7, *Campylobacter*, and *Listeria monocytogenes*, typically cause self-limiting gastroenteritis in humans, severe cases may necessitate antimicrobial treatment. Infections with MDR bacterial strains can lead to poor treatment responses, prolonged hospitalization, and increased mortality [63]. Additionally, food contaminated with resistant commensal and pathogenic organisms can transfer resistance genetic markers to other pathogenic and commensal organisms in the human gut, contributing to the spread of antibiotic resistance.

A recent global meta-analysis of antibiotic resistance in foodborne bacterial pathogens, such as *E. coli*, *Salmonella, Listeria monocytogenes*, and *Staphylococcus aureus*, revealed high rates of resistance to classes of antimicrobials, including aminoglycosides, β-lactams, chloramphenicol, fluoroquinolones, sulfonamides, and tetracyclines. Interestingly, bacterial pathogens isolated from food showed a higher rate of resistance compared to those isolated from humans for antibiotic classes such as aminoglycosides, β-lactams, chloramphenicol, sulfonamides, and tetracyclines in most of the foodborne pathogens [64,65]. This indicates that the selection for resistant populations of microorganisms occurs more frequently in the food chain than it does in humans, highlighting a high likelihood of the transfer of resistant from animals to humans through food.

Food-animal farms contribute to water pollution by releasing antimicrobial residuals as well as drug-resistant genes and bacteria. Runoff from farms, containing antibiotics and resistant bacteria, can enter nearby water bodies such as rivers, streams, and lakes. Clinically important bacteria such as *E. coli*, *Pseudomonas*, *Enterobacter*, *Shigella* spp., and *Vibrio* spp. have been reported in aquatic environments in Brazil [66]. These environments serve as crucial reservoirs of MDR bacteria, as demonstrated by Esposito et al. [67], who isolated two *Klebsiella pneumoniae* strains carrying the blaKPC-2 carbapenemase gene in aquatic environments in São Paulo, Brazil. This highlights the dissemination of certain strains beyond hospital settings, emphasizing the role of environmental reservoirs in the persistence and spread of highly resistant bacteria, potentially contributing to the amplification and spread of antibiotic resistance in the community. Similarly, Montenegro et al. [68] reported the occurrence of *K. pneumoniae* ST244 and ST11, extensively drug-resistant and producing KPC, NDM, and OXA-370, in wastewater in Brazil. Proper management of wastewater generated by farms, including adequate treatment before discharge or reuse, adherence to environmental regulations, and responsible agricultural practices, is essential to minimize adverse impacts on the environment and human health.

In Brazil, bacteria of significant medical importance, such as *Staphylococcus aureus*, *E. coli*, *Pseudomonas aeruginosa*, *Listeria monocytogenes*, *Salmonella* spp., *Shigella* spp., and *Vibrio* spp., have been detected in various samples obtained from animal farms and food sources. These samples range from cheese and milk to urban aquatic environments, hospital effluents, and shrimp farms [66]. Brazil, being a middle-income country and a major global food producer, faces the challenge of addressing antibiotic resistance on a large scale. This requires a comprehensive approach involving coordinated efforts from the government, healthcare sector, agriculture industry, and public awareness campaigns. Given Brazil’s role in significant food production for both domestic consumption and export markets, there is a need for a multifaceted strategy. Surveillance systems that monitor antibiotic-resistance prevalence in farm animals and environments are crucial for early detection and effective management. Additionally, promoting education and awareness among farmers, veterinarians, and consumers about the risks of antibiotic overuse and the importance of responsible practices is paramount.

It is crucial to recognize the role of bioaerosols in disseminating antibiotic-resistance genes and MDR bacteria. Bioaerosols released from farms can carry resistant bacteria over considerable distances, extending beyond the immediate farm surroundings and potentially impacting neighboring communities, wildlife, and environmental ecosystems. Studies have detected bioaerosols containing ARGs at distances of about 2 km from agricultural facilities, with potential dispersal footprints extending up to 10 km [69,70]. Despite being a crucial pathway for antibiotic-resistance dissemination, the airborne dispersal of resistant genes is an understudied phenomenon. This mechanism significantly contributes to the dispersion of highly diverse microbial communities, leading to shifts in the local abundance of resistant genes and alterations in the microbiota. Research has shown the presence of resistant genes in bioaerosols encoding resistance to various antibiotic classes, including aminoglycosides, beta-lactams, macrolides, quinolones, sulfonamides, tetracyclines, and vancomycin [71]. There is a pressing need for comprehensive research on bioaerosols, particularly in LMICs, to better understand their dynamics and potential implications.

In China, farmers often turn to herbal medicines as alternatives to antibiotics, drawing on the extensive history of traditional Chinese medicine. Despite the longstanding use of traditional medicine, herbal medicines must undergo thorough characterization and evaluation for safety, efficacy, and quality to be authorized as veterinary medicines. There are eleven guidelines providing the main framework for the registration process of herbal medicines. In 2014, Chinese herbal medicines for animals represented 15% of the total production value of the animal health industry, with a notable 12.39% increase in production value from the previous year. The growing support in China for herbal medicines and alternatives to antibiotics is evident, highlighted by various government reports issued in recent years [72].

In other LMICs, the use of antimicrobials in livestock is often inadequately regulated. This lack of regulation increases the likelihood of farmers freely using antimicrobials to treat their sick animals without proper veterinary diagnosis or understanding of the consequences. For instance, in Ethiopia, a recent study reported that antibiotic use in pastoral communities reached 86.7%, with tetracyclines, aminoglycosides, and trimethoprim-sulfonamides being the most commonly used. Notably, human-prescribed antibiotics (such as tetracyclines) were used for veterinary purposes by 18.5% of the study community. The study also revealed that a significant proportion of participants lacked knowledge about withdrawal periods for milk and meat after antibiotic treatment, and they reported administering higher doses than recommended [73]. Similarly, a recent study involving various livestock production systems in Kenya reported that over 90% of farmers used antibiotics, mainly for prophylactic purposes [74]. The uncontrolled and unregulated use of antibiotics in East African countries suggests that such practices could contribute significantly to the increasing problem of AMR, extending to bacterial foodborne pathogens affecting human health.

### 3.3. Implication of Knowledge, Attitude, and Practice of Farmers on Food Safety 

The interplay of farmers’ knowledge, attitudes, and practices holds significant implications for the broader landscape of food safety, shaping the foundation upon which agricultural processes and food production systems operate. In LMICs like Ethiopia and Kenya, small-scale farmers play a crucial role in producing most animal-origin foods such as milk [75,76]. Therefore, these farmers have a significant stake in ensuring the safety of food of animal origin. Their knowledge, attitude, and practices regarding the potential risk of pre-harvest farm-level activities on food safety are imperative for protecting consumers from unsafe foods.

Studies conducted in East Africa aiming to assess the level of knowledge, attitude, and practice (KAP) of dairy farmers have shown that farmers do not employ proper hygienic dairy practices, which has significant implications for the safety of milk produced under such conditions. Important biosecurity measures, such as cleaning sheds, washing cows’ udders and milkers’ hands, separating sick animals, and checking mastitis at the farm level, were not adequately implemented. The use of plastic milk containers is also a critical issue observed in Ethiopia and Kenya. Reports from these countries revealed that the large majority of smallholder dairy farmers use plastic containers for both milking and storage [75,77]. These plastic containers are more difficult to clean than aluminum containers and may be a potential source of milk contamination. Moreover, certain previous studies conducted in Africa have indicated an association between the use of plastic milk containers and contamination with coliform bacteria. These bacteria can proliferate rapidly in moist residues left in difficult-to-clean containers, emerging as significant sources of milk contamination [78,79].

An effective intervention for this issue could involve advocating the use of easily cleanable containers, such as aluminum vessels. Recognizing the financial constraints of many dairy farmers who may find aluminum containers unaffordable, an alternative approach could be to educate farmers about the potential risks associated with plastic milk containers. Emphasizing the importance of thorough cleaning using hot water and detergents can be a viable option to mitigate contamination risks and promote safer milk handling practices [77]. Furthermore, recent studies conducted in Ethiopia have highlighted specific practices such as cleaning pens, handwashing during milking, the type of milk containers used, udder washing, and overall farm management systems as risk factors for contamination with *E. coli* at both the farm level and in milk products [41,42]. Ensuring the safety of food of animal origin necessitates heightened awareness among farmers regarding the risks associated with the inadequate implementation of key biosecurity measures at the farm level.

In summary, the lack of farmers’ knowledge and adherence to proper biosecurity measures, coupled with the rational use of antimicrobials on farms, can markedly impact the safety of animal-source foods produced under such conditions.

## 4. Intervention Aiming Food Safety Problems at LMIC and Their Outcomes 

Implementing targeted interventions to address food safety challenges in LMICs is crucial, and exploring the outcomes of such interventions provides valuable insights into fostering safer food production and consumption practices worldwide. The diverse range of value chain actors, encompassing dairy farmers, milk traders, cooperatives, traditional milk processors, abattoir workers, public health officials, veterinarians, butcheries selling meats, and transporters, plays a crucial role in the food production process. Studies conducted in LMICs revealed that common challenges among some of these actors include a lack of good food handling practices that hinders compliance with recommended food safety and quality standards. Several factors contribute to this low compliance, including the absence of incentives for producing safe and high-quality products, inadequate infrastructure (such as road access, refrigerators, and transportation facilities), and insufficient enforcement of food safety laws and regulations [80]. Consequently, any intervention aiming to ensure food safety must strategically target these key areas.

While the above-mentioned value chain actors at different levels are known to play a critical role in mitigating risks on food safety, addressing farm-level food safety issues would be very important for LMICs for the following main two reasons. Firstly, the majority of foods are channeled to consumers through informal markets that are directly linked to farms. For example, more than 80% of animal products traded in sub-Saharan Africa go through informal markets, which lack organized sanitary inspection and frequently have unclear legal standing [25]. Secondly, most LMICs lack a well-functioning animal healthcare system; for example, in Ethiopia, a country that has the largest cattle population in Africa, the coverage and quality of veterinary services are less than satisfactory across the different livestock production systems [81]. Even so, the majority of small-scale farmers cannot afford to seek animal healthcare services. Additionally, the absence of a quality-based pricing system for animal products, coupled with prevalent informal market systems, may further dissuade farmers from seeking veterinary care for their sick animals. Consequently, the issue of sick animals persists as a critical food safety concern in many LMICs, where interventions addressing pre-harvest issues, such as animal health, are insufficient.

Implementing modern technologies to address food safety at the farm level can be challenging in LMICs. Nevertheless, simpler interventions, such as providing training to enhance the knowledge, attitude, and practices of farmers regarding the importance of maintaining good production practices, can be highly effective. A study conducted in Ethiopia that focused on milk contamination with *Staphylococcus aureus* found that implementing some simple post-harvest practices, such as washing milk containers with detergent and hot water, could reduce the risk of contamination with *S. aureus* by 66% [77]. Several reports have demonstrated how training small-scale farmers can enhance these practices, although the effectiveness of such interventions on a large-scale level is not extensively substantiated [7]. A targeted training intervention designed for women farmers in pastoral areas of Ethiopia, with the goal of enhancing knowledge, attitudes, and practice, demonstrated a significant improvement in KAP. However, a substantial proportion of participants still exhibited negative attitudes and incorrect practices post-intervention [82]. Numerous reports emphasize the positive impact of training on small-scale farmers to improve their practices. The key concept is that to achieve the desired positive outcomes, training should be complemented with interventions such as the introduction of locally adaptable technologies, incentivizing farmers for quality produce, and providing quality water and sanitation facilities.

Existing pre-harvest interventions can be broadly categorized into two groups. The first group aims to reduce specific bacterial pathogens from infecting farm animals. These interventions involve implementing external biosecurity measures meant to decrease the risk of introducing pathogens to a farm. Measures include animal breed selection, control of feed and water, and constructing physical barriers to restrict access. Internal biosecurity measures, such as strict hygienic practices, also play a role in preventing animal infection. The second group focuses on reducing bacterial pathogen concentration in farm animals. These interventions primarily aim to enhance host disease resistance, ultimately reducing pathogen load and minimizing the need for antibiotics, with the goal of pathogen reduction or elimination from farm animals [83].

The effectiveness of pre-harvest interventions in resource-limited countries has not been thoroughly evaluated, and existing laws, regulations, and interventions in LMICs often prioritize post-harvest aspects. Pre-harvest components are frequently left unaddressed in these regions. Farmers in LMICs generally comply with good agricultural practices when mandatory for export products, but evidence on the impact of such programs for domestic products is limited, suggesting less impact. Grace et al. [7] propose five critical factors for scalable and sustainable food safety interventions. The first factor is adherence to a gold standard design, such as a randomized controlled trial (RCT), for evaluation. Unfortunately, many intervention studies in LMICs often use simple pre- and post-intervention assessments that do not follow randomized control designs [84]. The second factor emphasizes the importance of the intervention’s environment, requiring support at both the micro and macro levels. An enabling environment includes acceptance by local authorities and consumers as well as the development of policies and regulations at higher levels. Interventions lacking this enabling environment may face sustainability challenges. Pre-harvest interventions in LMICs encounter scalability issues, partly due to the lack of an enabling policy environment and social acceptability. The third factor focuses on economic viability. In instances like Ethiopia, small-scale dairy farmers may struggle to afford items like aluminum milk containers. Many farmers opt for plastic jerry cans, which have no direct cost and are often repurposed oil containers found in households. The economic feasibility of interventions is crucial for scalability in resource-limited settings. The challenge of scalability in resource-limited settings is further highlighted by the reluctance or inability of small-scale farmers to afford higher-cost interventions, such as aluminum milk containers, despite their proven effectiveness. High-cost interventions often struggle to reach mass markets in LMICs, emphasizing the need to strike a balance between intervention cost and effectiveness. The fourth factor underscores the importance of instituting incentive systems to sustain effective interventions. While training can enhance knowledge, ensuring long-term changes in practice or behavior may require additional motivators. Incentives play a crucial role in bridging the gap between knowledge and real-life application. However, establishing incentive mechanisms in LMICs is complex, as producers may be unwilling to invest in food safety processes, and consumers may resist paying a premium for safer food.

In summary, the adoption of pre-harvest measures in LMICs is not widespread. Even in some developed countries, there is often a greater emphasis on post-harvest measures compared to pre-harvest strategies. For example, in the USA, efforts to intensify post-harvest measures aim to reduce *Salmonella* contamination in meat, while pre-harvest control strategies have not been developed and implemented at an equivalent pace [85]. In contrast, in European countries, both pre-harvest and post-harvest strategies are implemented to reduce the risk of food contamination such as the efforts used for *Campylobacter* in broiler chicken production [86]. 

## 5. One Health Approach: A Key Strategy for Mitigating Pre-Harvest Food Safety Risks 

The interconnectedness of human health with animals, the environment, and the ecosystem has gained recognition in contemporary academia and the public health arena. The ongoing global trend of rapid human population growth, coupled with associated environmental degradation, contributes to the complexity of the present and future public health threats faced by humanity.

Interestingly, our food system and the environment are believed to have played a significant role in the recent emergence of diseases such as Middle East respiratory syndrome (MERS), Ebola, and avian influenza [87]. Moreover, food production and livestock systems are known to contribute to the escalating risk of the emergence of new zoonotic diseases. The prevailing notion is that mitigating such threats necessitates the involvement of multidisciplinary and multi-sectoral actors across the human, animal, and environmental spectra. Therefore, many low- and middle-income countries (LMICs) are making varying degrees of efforts towards implementing the One Health approach to address complex health issues at the human–animal interface and their shared environment.

China has made efforts to tackle zoonotic diseases through policies, funds, and technologies. It has enacted the Infectious Diseases Prevention Law and established special programs for high-risk zoonotic diseases. The average annual growth rate of the Chinese government’s budget for major infectious diseases between 2016 and 2020 was 3.31%, excluding the budget for COVID-19. In addition to promulgating and enhancing the Wildlife Protection Law and the Animal Epidemic Prevention Law, China has also enacted surveillance systems for both livestock and wildlife to improve animal health [88]. Moreover, China has been addressing food security issues, including reduced arable land, growing food demand, and insufficient food surveillance. Policies such as the National Agricultural Sustainable Development Plan (2015–2030) and the Food Safety Law have been implemented, encouraging strategies like returning crop residues to fields, utilizing organic fertilizers, and strict surveillance of food additive use. To reduce the massive amount of food waste, a national “clean your plate” promotion has been in place since 2013, followed by the Anti-Food Waste Law in 2021 [88]. Furthermore, China has put extensive effort into controlling antibiotic use, resulting in a reduction in antibiotic usage among hospitalized patients from 59.40% in 2011 to 36.0% in 2019. This achievement followed years of controlling clinical antibiotics between 2011 and 2013. In 2014, China implemented the National Action Plan to Contain Antimicrobial Resistance, which was the first to address antibiotic-resistance problems from a holistic perspective. It stressed the importance of regulations in healthcare and agriculture, intersectoral collaboration with clear responsibilities, and management of environmental pollution [88].

The One Health approach advocates for the implementation of a multisectoral system encompassing human, animal, and environmental health. It fosters multidisciplinary collaboration among global, national, and sub-national institutions to achieve optimal health for people, animals, and the environment by building capacities at all levels. This comprehensive approach to global public health threats facilitates prevention, early detection, and effective response mechanisms.

Enhancing the value of the China–Africa “One Health” strategy involves creating partnerships, developing frameworks, and building capacity through the utilization of existing and innovative China–Africa health initiatives. Additionally, mobilizing efforts to address climate change, food safety, disaster mitigation, and lifestyle adaptations is crucial in combating both emerging and current infectious disease threats. Establishing an epidemic surveillance-response system requires a commitment to global collaborative coordination and sustainable financing mechanisms. Leveraging current and forward-looking strategies is essential to creating a resilient system that can effectively respond to the evolving challenges posed by infectious diseases on a global scale [89].

Many African countries are considering the One Health approach as a guiding principle for mitigating zoonotic diseases, despite the significant gap in institutionalizing One Health. For example, Ethiopia has adopted the One Health approach to respond to existing and future threats as part of the execution of the Global Health Security Agenda. The key achievements obtained so far include the establishment of (i) national One Health Steering Committee and Technical Working Groups, (ii) prioritization of zoonotic diseases based on their impact on humans and livestock, (iii) development of prevention and control working documents for prioritized zoonotic diseases, (iv) joint disease surveillance and outbreak investigation, (v) prioritization of zoonotic diseases, capacity building, and (vi) promotional work on the One Health approach in the country [90]. Nonetheless, the implementation of the One Health approach among different stakeholders still faces several challenges, such as lack of proper coordination among the One Health actors, unavailability of One Health-trained workforce, lack of funds, and, most importantly, One Health governance is based on ad hoc commitment lacking proper institutionalization. Additionally, while the national One Health steering committee has a number of technical working groups for major zoonotic public health threats, it lacks working groups that address food-safety-related public health issues. It is believed that effectively implementing the One Health approach to address food safety issues enables countries to timely detect, prevent, and respond to emerging and re-emerging foodborne diseases [87].

The national food safety strategy proposed by China lays the foundation for a unique Chinese framework in the food safety management system, with the core goal of ensuring its people “eat at ease and safely”. The strategy adopts a comprehensive approach covering the entire food chain, starting from animal feed production and extending to the consumption of the final product by consumers. Consistent oversight of the food chain will be adopted, ensuring equal attention is given to all risks. There will be no varying levels of safety; both domestically consumed and exported food will adhere to identical food safety standards [88].

Under China’s state-level guiding principle of “integrated marketing, supervision, industry, and management,” the strategy focuses on harmonizing the domestic food market, optimizing government supervision, promoting high-quality development, and coordinated social governance. Additionally, this strategy is supported with increased financial investment, education, and related regulations. Specific measures include: (1) establishing a unified, modern, open market system with managed competition, (2) promoting optimization, collaboration, and efficiency of the supervision system, and (3) establishing a social governance model based on collaboration, participation, and common interests. While applying the WHO strategy for food safety, the Chinese government has proposed its own timetable and roadmap for its domestic food safety strategy, including (1) zero tolerance of systemic food safety risks by 2020 and constantly improving the level of food safety assurance, (2) establishing a strict, highly efficient, and socially-governed food safety governance system by 2027, (3) achieving the modernization of food safety governance by 2035, and (4) achieving universal modernization of food safety governance and approaching the world’s top food-safety-level ranking by 2050 [87].

Effectively implementing the One Health approach at the farm level is believed to have the potential to improve global health by establishing best practices for producers. For instance, in dairy farm production systems, the use of high-quality water, feed, and supplies, along with instituting protocols, is expected to enable the prevention of the spread of foodborne diseases and the use of chemical adulterants. Moreover, a proper disposal system for waste generated at farms should be implemented to avoid environmental contamination [91].

Little information is available in LMICs regarding the waste disposal system at the farm level and how the waste affects both human health and the ecosystem in general. In addition, farms often lack appropriate biosecurity measures. For example, a recent study from Ethiopia that aimed to assess biosecurity measures in dairy farms revealed that the majority of the farms missed essential biosecurity measures such as footbaths at their gate points, isolation areas for either sick or newly introduced cattle, and checking the health status or quarantining newly introduced cattle. The use of personal protective equipment was also reported to be very low [92]. This study concluded that dairy farms in central Ethiopia are characterized by a lack of satisfactory biosecurity measures and hence require intervention with the view of improving animal health and averting potential public health threats. Such interventions inherently necessitate a central focus on the One Health approach.

## 6. Conclusions

In LMICs, the rising demand for animal-source food is closely tied to the increasing population. However, ensuring the production of safe animal-origin food remains a major challenge. While various countries tend to have some form of policy and regulatory framework to address food safety at the post-harvest level, primarily focusing on processed foods, most legislation and regulations in LMICs fall short in addressing food safety issues at the farm level (pre-harvest). They predominantly concentrate on formal markets, overlooking the large majority of animal-origin food marketed through informal markets, which are not adequately regulated. The large majority of food of animal origin in LMICs is marketed through informal markets that are overlooked by the available regulatory system. Considering their significant role in food safety, it is imperative to make informal markets part of the food safety regulatory system. Additionally, some food safety policies in LMICs do not effectively resolve mandate conflicts among different governmental bodies, leading to confusion and a lack of accountability regarding responsibilities along the value chain.

Several critical food safety issues at the farm level persist in LMICs. These include the absence of veterinary services for sick animals, low adherence to good farming practices (such as inadequate hygiene), and the use of inappropriate farm equipment, like plastic milk containers, believed to contribute to contamination. Antibiotic-resistance foodborne pathogens are prevalent in foods of animal origin and the farm environment in LMICs, partially due to the widespread use of antimicrobials. Establishing AMR surveillance systems that monitor antibiotic-resistant bacterial prevalence in farm animals and environments is essential. Moreover, there is a significant gap in promoting education and awareness among farmers, veterinarians, and consumers about the risks of antibiotic overuse and the importance of responsible practices in LMICs. 

Numerous pre-harvest interventions have been implemented to mitigate risks associated with food of animal origin in LMICs. However, despite their effectiveness at a small scale, these interventions often face challenges in scalability to larger domains. Primarily centered on training farmers, these interventions tend to have short-lived impacts, necessitating complementary measures such as the provision of technologies and incentives. Farmers commonly adhere to good agricultural practices when mandated for export products, but the evidence regarding the impact of such programs on domestic products is limited, suggesting that these interventions may yield less impact in domestic contexts.

While some LMICs are making efforts to implement the One Health approach to prevent zoonotic disease outbreaks, this framework is still in its infancy in terms of institutionalization and proper coordination among multidisciplinary and multisectoral actors. Foodborne-disease outbreaks inherently require addressing the problem within a One Health framework. However, the practice of tackling food safety issues from a One Health perspective is notably limited in LMICs.

## 7. Way Forward 

Given that a significant portion of animal-source foods in most LMICs flows through informal markets, it is essential for food safety legislation, policies, and regulatory frameworks to consider this context. The predominant intervention in LMICs is training, focusing on enhancing the knowledge, attitude, and practices of farmers. Although training can lead to temporary behavioral changes, other interventions, such as establishing systems for quality-based pricing and providing affordable technologies to farmers, are crucial for ensuring the sustainability of these efforts. Interventions aimed at reducing foodborne pathogens at the farm level should be scalable, sustainable, and cost-effective. Advocating for a One Health approach is imperative for the proper implementation of pre-harvest interventions at the farm level. This involves addressing food safety from both environmental and animal health perspectives, emphasizing a holistic and integrated approach.

## Figures and Tables

**Table 1 animals-14-00786-t001:** Summary of major bacterial foodborne pathogens reported in LMICs from animal-origin foods and farm environments.

Bacterial Species	Sample Source	Prevalence (%)	References
*E. coli* O157:H7	Milk	4.5	Mesele et al., 2023 [41]
	Water	8	
	Manure	4	
	Feces	3.9	
	Milk and milk products	0.2	Sarba et al., 2023 [42]
	Cattle feces	0.7	Su et al., 2021 [45]
	Rectal swab	1	
	Carcass swab	4	
	Processed milk	2.04	Monique et al., 2016 [46]
	milk	2.2	Kang’ethe et al., 2007 [47]
	Cattle feces	5.2	
*S. aureus*	Cows’ udder	28.1	Abebe et al., 2023 [43]
	Cow milk	61.7	Kou et al., 2021 [48]
	Milk and milk products	10.7	Gebremedhin et al., 2022 [49]
	Milk	10.9	Dittmann et al., 2017 [50]
*Listeria monocytogenes*	Milk	2	Gume et al., 2023 [51]
	Milk and milk products	5.6	Seyoum et al., 2015 [52]
	milk	0.36	Ning et al., 2013 [53]
*Salmonella*	Milk	10.4	Alemayehu et al., 2022 [54]
	Milk	20.5	Castañeda-Salazar et al., 2021 [55]
	Cattle feces	2.3	Eguale et al., 2016 [56]
	Meat carcass	2.5	Ketema et al., 2018 [57]
	Cattle feces	4.1	

## Data Availability

Data are contained within the article.

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
