# Peer review of "Pre-Harvest Food Safety Challenges in Food-Animal Production in Low- and Middle-Income Countries"

_animals, 2024, doi:10.3390/ani14050786_

Round 1
Reviewer 1 Report
Comments and Suggestions for Authors
Overall, the authors have done a great job expanding the existing knowledge of food safety issues within Low and Middle Income Countries. Information presented is well written and formatted within the requirement of the authors guidelines of the Journal. A few specific comments per line item are listed below.
Line 4: Formatting of superscripts and commas identifying authors needs review and corrected.
Line 121-122: Formatting, delete extra space.
Line 122-129: Body the text appears in Bold Font, authors should consider returning font to authors guidelines as specified by the journal. Appears that there is a period or formatting of the final thought on this section by the authors.
Line 129-130: Delete the extra space.
Line 164-170: Could the authors consider adding a reference regarding these food safety laws described for potential readers?
Line 220: Spelling of citizens???? Please confirm netizens is presented accurately.
Line 254: Figure 1 should appear in the body of text as close to the first mention as possible. Would recommend that the authors consider placing Figure 1 at or near line 206.
Table 1. Review and correct presentation of reference Mesele et al. 2023…..
Table 1. Review and correct presentation of reference Su et al., 2021…..
Table 1. Review and confirm the font for all text is the same size. It appears that there are some references that the font size appears larger than the other references.
Line 597: Delete
Line 746: Please review the author contributions, it appears there is one co-author missing?
Author Response
Overall, the authors have done a great job expanding the existing knowledge of food safety issues within Low and Middle Income Countries. Information presented is well written and formatted within the requirement of the authors guidelines of the Journal. A few specific comments per line item are listed below.
Response: Thank you for your encouraging feedback.
Line 4: Formatting of superscripts and commas identifying authors needs review and corrected.
Response: These have been corrected.
Line 121-122: Formatting, delete extra space.
Response: Change has made in the text.
Line 122-129: Body the text appears in Bold Font, authors should consider returning font to authors guidelines as specified by the journal. Appears that there is a period or formatting of the final thought on this section by the authors.
Response: Change has been made in the text.
Line 129-130: Delete the extra space.
Response: Change has made in the text.
Line 164-170: Could the authors consider adding a reference regarding these food safety laws described for potential readers?
Response: We have added four references to support the food safety laws in China.
“Furthermore, in China, the government has introduced several plans to enhance food safety, including the "Food Safety Law of the People's Republic of China," the "Animal Epidemic Prevention Law of the People's Republic of China," the "Regulations on the Management of Pig Slaughtering," and the "Three Year Action Plan for Strict Standards to Promote and Improve Safety in Livestock and Poultry Slaughtering." [19-22]. These plans aim to regulate meat safety across various stages, from animal breeding and slaughter to distribution.”
Line 220: Spelling of citizens???? Please confirm netizens is presented accurately.
Response: The word “netizens” meant those who are internet users.
Line 254: Figure 1 should appear in the body of text as close to the first mention as possible. Would recommend that the authors consider placing Figure 1 at or near line 206.
Table 1. Review and correct presentation of reference Mesele et al. 2023.
Response: Thank you for this important comment, we have corrected the reference.
Table 1. Review and correct presentation of reference Su et al., 2021.
Response: We double checked the reference and it is correctly presented as indicated in the list.
Table 1. Review and confirm the font for all text is the same size. It appears that there are some references that the font size appears larger than the other references.
Response: Changes have been made to ensure font for all text is the same.
Line 597: Delete
Response: Change has been made.
Line 746: Please review the author contributions, it appears there is one co-author missing?
Response: The co-author was added to the author contributions.
Reviewer 2 Report
Comments and Suggestions for Authors
The review is interesting and very timely and deals with a topic that needs more and more global attention. However, some changes are needed below.
Introduction and title
The review focuses mainly on foodborne zoonoses, but biohazards also include parasites (zoonotic and non-zoonotic) that have not been treated. There are also problems with drugs other than antibiotics and other chemical hazards (e.g., mycotoxins, pesticides, etc.); these were also not discussed.
The scope of the review should be further refined to make it more consistent with what is described.
In addition, it might be helpful to specify more clearly which are the main countries analyzed.
Finally, the title should also be changed to make the document even more specific and interesting.
L. 128-129: check the space, please.
L. 274-279: animal welfare, animal cleaning? Are they not important or are they considered in management?
L.577-581. Instead, in Europe, the strategy implemented is preventive working on animal farms.
Author Response
The review is interesting and very timely and deals with a topic that needs more and more global attention. However, some changes are needed below.
Introduction and title
The review focuses mainly on foodborne zoonoses, but biohazards also include parasites (zoonotic and non-zoonotic) that have not been treated. There are also problems with drugs other than antibiotics and other chemical hazards (e.g., mycotoxins, pesticides, etc.); these were also not discussed.
Response: We have made changes to clarify that the review focuses on foodborne bacterial zoonotic pathogens. Regarding the drugs, our focus is on administrated antibiotics as they relate to development and spread of antibiotic resistant bacteria.
The scope of the review should be further refined to make it more consistent with what is described.
Response: Changes have been made in the manuscript text to reflect the scope of the review.
In addition, it might be helpful to specify more clearly which are the main countries analyzed.
Response: The review addresses preharvest food safety issues in LIMCs in general and provide examples from specific countries/regions, mainly China, Ethiopia, Brazil, and Middle East. This has been clarified in the introduction.
Finally, the title should also be changed to make the document even more specific and interesting.
Response: The title has been modified as follows “Pre-Harvest Food Safety Challenges in in Food Animal Production in Low and Middle-Income Countries”
- 128-129: check the space, please.
Response: Change has been made to remove extra space.
- 274-279: animal welfare, animal cleaning? Are they not important or are they considered in management?
Response: Animal welfare and animal cleanness are important preventive actions at the farm level to reduce the risk of food contamination. We have added them to the text.
L.577-581. Instead, in Europe, the strategy implemented is preventive working on animal farms.
Response: We have added a text to briefly mention that European countries follow preventive strategy at the farm level.
Round 2
Reviewer 2 Report
Comments and Suggestions for Authors
The manuscript has been sufficiently improved.